# Reactive Collision Avoidance of an Unmanned Surface Vehicle through Gaussian Mixture Model-Based Online Mapping

**Dongwoo Lee**  **and Joohyun Woo ***

Department of Naval Architecture and Ocean Systems Engineering, Korea Maritime and Ocean University, Busan 49112, Korea; leedw0426@g.kmou.ac.kr
* Correspondence: jhwoo@kmou.ac.kr

**Abstract:** With active research being conducted on maritime autonomous surface ships, it is becoming increasingly necessary to ensure the safety of unmanned surface vehicles (USVs). In this context, a key task is to correct their paths to avoid obstacles. This paper proposes a reactive collision avoidance algorithm to ensure the safety of USVs against obstacles. A global map is represented using a Gaussian mixture model, formulated using the expectation–maximization algorithm. Motion primitives are used to predict collision events and modify the USV's trajectory. In addition, a controller for the target vessel is designed. Mapping is performed to demonstrate that the USV can implement the necessary avoidance maneuvers to prevent collisions with obstacles. The proposed method is validated by conducting collision avoidance simulations and autonomous navigation field tests with a small-scale autonomous surface vehicle (ASV) platform. Results indicate that the ASV can successfully avoid obstacles while following its trajectory.

**Keywords:** gaussian mixture model (GMM); reactive collision avoidance; motion primitives; unmanned surface vehicle (USV); robot operating system (ROS); simulation



## 1. Introduction

In the fourth industrial revolution era, characterized by big data, artificial intelligence, and autonomous driving, technologies in all domains are being rapidly advanced through innovation and fusion. At present, unmanned transport vehicles are being actively researched in the fields of automobiles and aviation. Furthermore, in the shipbuilding and offshore domains, the realization of vessel autonomy, remote control systems, and fault diagnosis systems is being focused on, according to the global trend to reduce human intervention. An unmanned surface vehicle (USV) refers to a ship that is operated remotely or automatically without an active crew. Unmanned ships can perform dangerous missions such as reconnaissance patrol and mine clearing. Furthermore, in the private sector, such ships can be used to realize passenger transport, coast guarding, marine exploration, and rescue. In such applications, cost effectiveness and work efficiency can be enhanced through the automation of ships, and the number of casualties can be minimized by preventing human intervention.

A key technology for unmanned ships is collision avoidance, aimed at ensuring the safety of a sailing vessel while maintaining its course when the vessel encounters other ships or obstacles. Research on collision avoidance technology has been performed since before the development of USV, and many collision avoidance techniques have been established. To determine the collision risk of a ship, a ship domain can be introduced using the relative distance between the ship and other ships [1,2]. Moreover, AIS data and probabilistic approaches can be applied [3–7]. According to an alternative theory, a moving robot can be expressed as a dynamic obstacle [8–10]. Certain methods for collision avoidance use the direction priority sequential selection algorithm; International Regulations for Preventing Collisions at Sea (COLREGs), defined by the International Maritime Organization [11];

and virtual vector fields [12]. In terms of the control method, collision avoidance methods can be divided into deliberative and reactive control methods. Although deliberative control methods exhibit a high performance in tasks, their reaction speed is low, and these methods cannot effectively adapt to environmental changes. In contrast, although reactive control can promptly respond to changes in the external environment, accurate control is challenging to realize. Overall, although various methods can be used for identifying the collision risk of a vessel and performing collision avoidance, the vessel may not be able to promptly respond to unfamiliar obstacles or obstacles that suddenly appear during a voyage. In such situations, reactive collision avoidance methods are highly effective. In particular, in very urgent situations, the vessel may maneuver in a way that does not comply with COLREGs to avoid a collision situation. Coping methods in emergency situations are described in Rule 2(b) of COLREGs [13]. Several researchers have attempted to realize the collision avoidance of autonomous surface vehicles (ASVs) and unmanned aerial vehicles by using the control barrier function [14] and real-time local Gaussian Mixture Model (GMM) maps [15]. A commonly used technique for local collision avoidance is based on a conceptual force known as the potential field, which is exerted by the obstacle on a robot [16]. This concept has been extended to propose the virtual force field (VFF) concept [12]. Notably, in these methods, a local minima occurs in the process of force composition. Consequently, the robot cannot follow its trajectory or traverse a narrow path. In this context, it is necessary to establish a simple method that can allow a vessel to follow its trajectory while effectively avoiding obstacles.

Considering these aspects, this study is focused on establishing a reaction control-based system to ensure the safety of an unmanned ship. Mapping is performed through the GMM by using the LiDAR information of an unmanned ship, and collision avoidance is performed in a reactive manner.

The remaining paper is organized as follows. Section 2 describes the global mapping technique based on GMM to create a map of the surrounding environment of a USV. Section 3 describes the proposed reactive collision avoidance method to realize real-time collision avoidance based on the GMM-based global map. Section 4 describes the reactive collision avoidance simulations performed using the proposed technique to evaluate its performance. Section 5 presents the conclusions.

## 2. GMM-Based Mapping

### 2.1. GMM

The GMM is used to detect obstacles and express them on the global map. In the GMM approach, an irregular distribution is expressed as several normal distributions (Gaussian distributions). In a one-dimensional environment, it is assumed that the data points are distributed on a straight axis. The points correspond to a normal distribution, which can be characterized by parameters such as the mean ($\mu$), standard deviation ($\delta$), and weight ($\pi$). The GMM refers to the model in which these Gaussian distributions are linearly combined. If the data are expanded in two dimensions, a GMM with a three-dimensional shape is created, and an ellipse can be obtained when a part of the GMM is projected onto a two-dimensional plane. Figure 1a shows the case in which points are distributed in one dimension, and the corresponding GMM is expressed as a graph. Figure 1b shows an example of a 3D GMM corresponding to 2D data.

GMM has the advantage of having a fast calculation speed while expressing obstacle information with a probabilistic distribution. In addition, if the number of components and parameter values are properly provided, accurate modeling is possible. However, as a disadvantage, it requires advance information on the encounter environment to determine parameters for the algorithm, and local maxima problems can arise.

The probability that a data point $\mathbf{x} \in \mathbb{R}^2$ belongs to a specific Gaussian distribution is defined as Equation (1) [17]. $M$ is the number of Gaussian distributions. Generally, the

Gaussian distribution is determined by the parameters $\mu$, covariance ($\Sigma$), and $\pi$. The GMM is defined by a set of parameters $\Theta = \{\mu_1, \mu_2, ..., \mu_M, \Sigma_1, \Sigma_2, ..., \Sigma_M, \pi_1, \pi_2, ..., \pi_M\}$.

$$p(\mathbf{x}|\Theta) = \sum_{k=1}^{M} \pi_k N(\mathbf{x}|\mu_k, \Sigma_k) \tag{1}$$

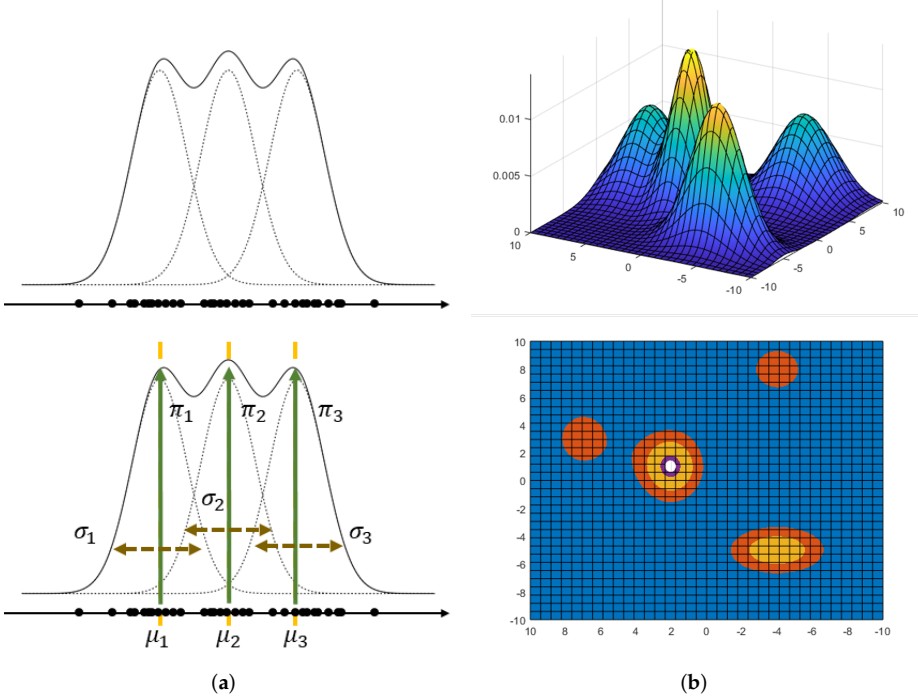

**Figure 1.** Examples of GMM. (**a**) GMM based on 1D data; (**b**) GMM and ellipses based on 2D data.

The expectation–maximization (EM) algorithm is used to estimate the unknown parameter ($\Theta$) [17–19]. The EM algorithm consists of two steps:

- E-step
  The data are labeled using the given parameters. In the initial stage, random parameters are assigned because the parameters are not known.
- M-step
  The parameters of each group are recalculated through maximum likelihood estimation (MLE). In general, the MLE method can be used to obtain the parameters of a random variable by sampling a random variable. The likelihood is expressed as in Equation (2).

$$L(\Theta|\mathbf{x}) = \prod_{k=1}^{n} P(\mathbf{x}_k|\Theta) \tag{2}$$

Specifically, the likelihood is calculated by multiplying a given parameter by all probability density values of n data. $L(\Theta|\mathbf{x})$ means the degree to which the parameter is consistent with the observed data $\mathbf{x}$. The right-hand side of the equation means multiplying all the probabilities of the data $\mathbf{x}$ for a specific parameter. The resulting expression is termed the likelihood function. For ease of calculation, the log-likelihood function is used, as defined in Equation (3).

$$\ln L(\Theta|\mathbf{x}) = \sum_{k=1}^{n} \ln P(\mathbf{x}_k|\Theta) \tag{3}$$

The MLE function aims to maximize the likelihood. To identify the maximum value, partial differentiation is performed with respect to $\Theta$, and the $\Theta$ that leads to a zero value is determined, as indicated in Equation (4).

$$\frac{\partial}{\partial\Theta}L(\Theta|\mathbf{x}) = \sum_{k=1}^{n}\frac{\partial}{\partial\Theta}\ln P(\mathbf{x}_k|\Theta) = 0 \tag{4}$$

The E-step and M-step are repeated until the distribution converges to yield the optimal parameter.

### 2.2. Global Map Creation

The discharge map is generated as follows. First, the global map in which obstacles are registered is initialized. Subsequently, point cloud information is received from the 3D LiDAR. The points only have raw information that is not clustered, and the reference coordinate system of the points is the object coordinate system fixed to the ship. If GMM fitting is performed with the points received through the EM algorithm, the confidence ellipse corresponding to each distribution can be obtained.

In the $xy$ coordinate system, the equation of an ellipse with major and minor axes $a$ and $b$ ($a \geq b$), respectively, parallel to the coordinate axes, can be expressed as in Equation (5).

$$\left(\frac{x}{a}\right)^2 + \left(\frac{x}{b}\right)^2 = 1 \tag{5}$$

In this case, the axis of the confidence ellipse can be expressed by scale factor $s$ and the standard deviations $(\sigma_x, \sigma_y)$, as indicated in Equation (6).

$$\left(\frac{x}{\sigma_x}\right)^2 + \left(\frac{y}{\sigma_y}\right)^2 = s \tag{6}$$

$s$ is determined by the confidence level, and the sum of the squares of the Gaussian distribution data corresponds to the chi-square distribution. $s$ can be easily obtained using the chi-square table [20]. In the GMM, the confidence ellipse is determined by the mean and covariance of the Gaussian distribution. Figure 2 shows the confidence ellipse determined by the mean and covariance. The mean of the GMM determines the center of the ellipse, and the covariance determines the size of the ellipse. $\mu_1$ and $\mu_2$ are the mean values of the GMM, which become the center of the ellipse. $\lambda$ is the Eigenvalue of the covariance ($\lambda_1$ and $\lambda_2$ are Eigenvalues corresponding to the semi-major axis and the semi-minor axis, respectively), and $\alpha = \tan^{-1}\left(\frac{\mathbf{v}_{1x}}{\mathbf{v}_{2y}}\right)$ is the angle between the semi-major axis of the ellipse and x-axis ($\mathbf{v}_1$ is the Eigenvector about the major axis).

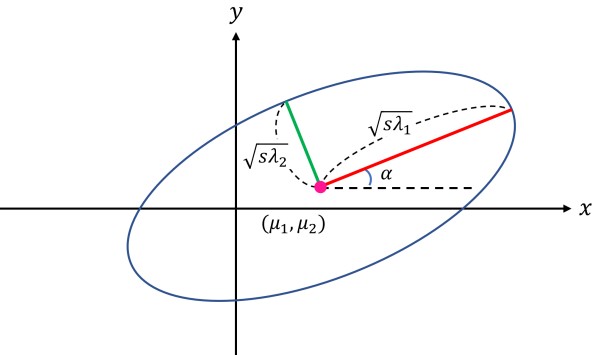

**Figure 2.** Confidence ellipse derived from the Gaussian distribution.

After drawing ellipses for each Gaussian distribution element, the information of the obstacles is expressed on the global map. The process of updating the map is illustrated

in Figure 3. All the registered ellipses have position and size information and update time information. $d_{GMM}$ denotes the distance between each GMM, corresponding to the Euclidean distance. $t_{GMM}$ represents the time at which the GMM is updated. Because GMM fitting is performed by the point cloud received in real time, it can be newly registered even for obstacles that have already been mapped. To prevent this problem, if the distance between two ellipses ($d_{GMM}$) is less than 1 m, the other ellipse is regarded as the same obstacle and update is not performed. Instead, the update time ($t_{GMM}$) of the previously registered obstacle is updated. In addition, obstacles registered on the global map are designed to disappear after a certain period because the obstacles are considered to be dynamic. In particular, if a registered obstacle is continuously maintained, the obstacle will continue to remain on the path through which the dynamic obstacle has passed. In this case, proper obstacle map expression and collision avoidance cannot be realized.

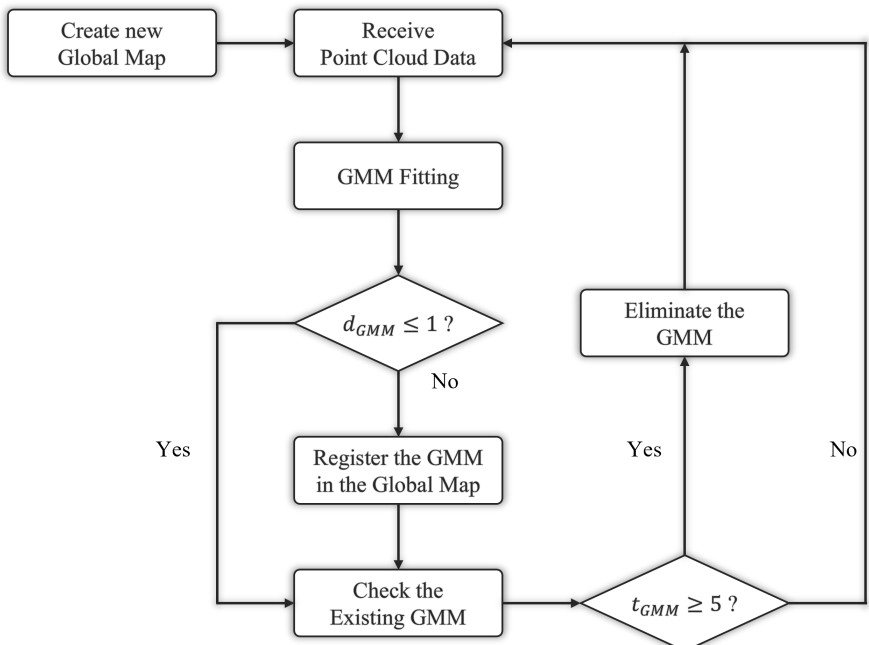

**Figure 3.** Process flow of global map update.

## 3. Reactive Collision Avoidance

### 3.1. Reactive Collision Avoidance Algorithm

The USV identifies the target point by following the target psi through the pure pursuit guidance algorithm. That algorithm is one of the ways of generating guidance commands so that the bow of the USV always aims for the target point. A detailed explanation is given in Figure 4. The state of an object can be expressed by three components representing the position ($X_{os}$, $Y_{os}$) and heading angle ($\psi_{os}$) and two components representing the surge speed ($v_x$) and angular speed ($\omega$). Assuming that $X_{goal}$ and $Y_{goal}$ are the position of the destination, the angle between the X-axis and the destination is defined as Equation (7).

$$\psi_d = \tan^{-1}\left(\frac{Y_{goal} - Y_{os}}{X_{goal} - X_{os}}\right) \tag{7}$$

The algorithm aims to make the error of angle $\psi_{os}$ and angle $\psi_d$ zero. The USV uses the error to input a control command. At this time, a PD controller is used.

$$\psi_{err} = \psi_{os} - \psi_d \tag{8}$$

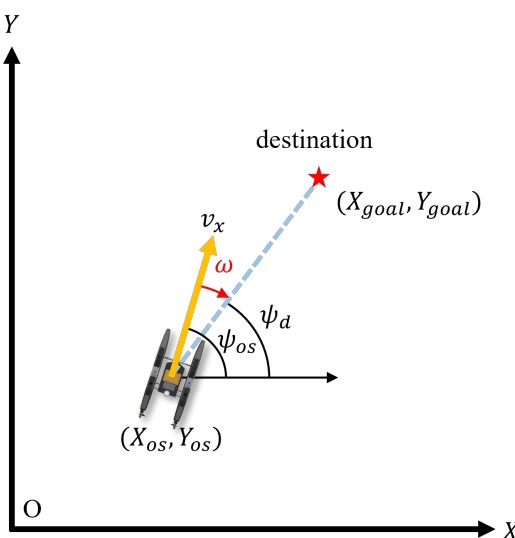

**Figure 4.** Coordinate system of the USV.

If there is no risk of collision, waypoint tracking is performed as above, but when an obstacle is detected, the control of the USV is performed in a different way. Collision avoidance of the USV is implemented as follows. The USV has a virtual local trajectory list based on $n$ yawrates, defined by motion primitives. Therefore, the motion of the vessel pertaining to each yawrate can be predicted. While the USV is moving forward, local trajectories that do not collide with obstacles are determined from numerous motion primitives, and obstacle avoidance and path following is realized by adopting the yawrate that most accurately follows the desired psi. Figure 5 shows the procedure by which the USV avoid collisions. Finally, the yawrate is entered as a steering command of the ship.

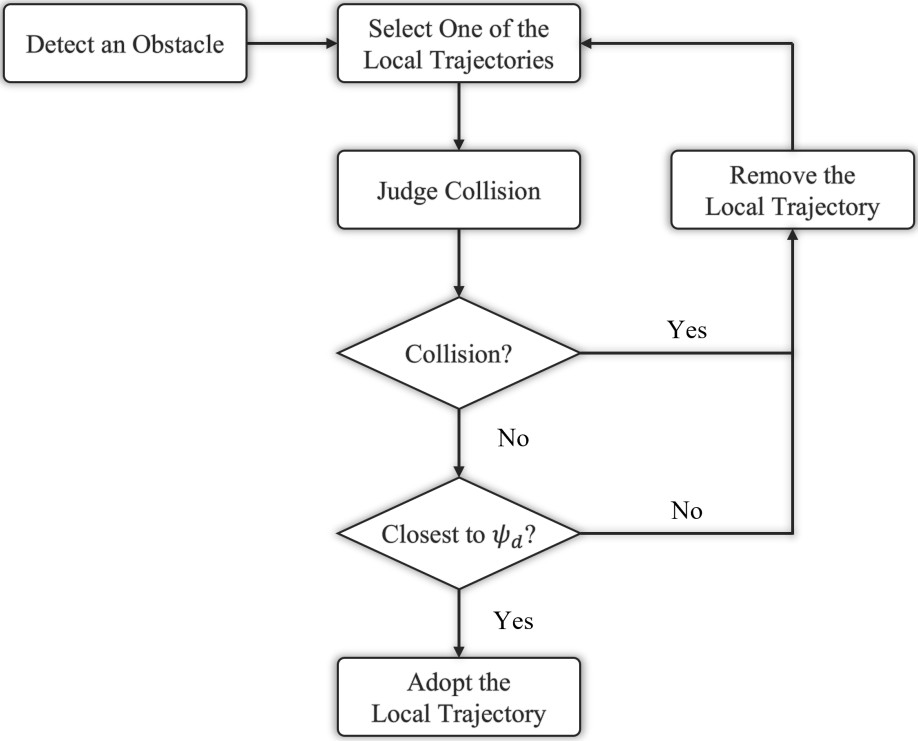

**Figure 5.** Procedure of collision avoidance check and motion primitive adoption.

### 3.2. Motion Primitives

Motion primitives are motions of an object calculated in advance to predict the movement path of the vessel according to the actuator command based on the current position of the vessel. In certain cases, motion primitives have been used for collision avoidance of autonomous ground vehicles (AGVs) [21] or vessel path planning [22]. In this study, the ship behavior is predicted using the unicycle model. The unicycle model is a kinematic model used to simply express the motion of an object moving on a two-dimensional plane. In the model, the motion of an object is expressed only with forward speed ($v_x$) and heading angle ($\psi$). The representative variables related to the unicycle model are expressed in Figure 6. The motion primitives of the unicycle model can be approximated as in Equation (9) [23]. In particular, motion primitives are paths created by accumulating vessel positions at time $t \in [0, T]$. T refers to a specific elapsed time, used to express the trajectory. $\mathbf{x}_t = (x_t, y_t, \psi_t)$ represents the position of the USV at a specific time, and $\mathbf{x}_0$ is the initial position.

$$\mathbf{x}_t = \mathbf{x}_0 + \begin{bmatrix} \frac{v_x}{\omega}(sin(\omega t + \theta) - sin(\theta)) \\ \frac{v_x}{\omega}(cos(\theta) - cos(\omega t + \theta)) \\ \omega t \end{bmatrix} \tag{9}$$

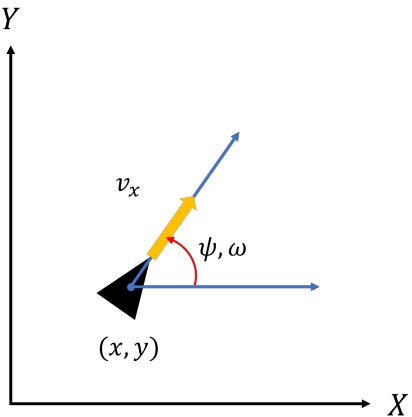

**Figure 6.** Unicycle model.

Motion primitives have $K \times N$ discretized points corresponding to $K \in [0, T]$ time parameters and $N$ yawrate variables, as shown in Figure 7. Each point is compared with $M$ Gaussian distribution elements to determine whether a collision may occur. For discrete motion primitives, if the left-hand side of Equation (6) is smaller than the right-hand side, the point is considered to collide with the obstacle because the point exists in the circle.

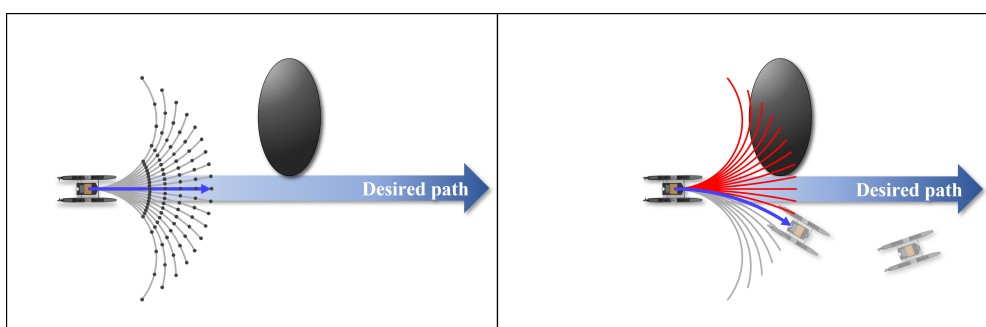

**Figure 7.** Schematic process of collision detection and collision avoidance.

Algorithm 1 describes the process of judging whether a collision occurs, as a pseudocode. $\mu_{mx}$ and $\mu_{my}$ are the mean values of the $m$th Gaussian distribution elements, that is, the $x$ and $y$ coordinates of the confidence ellipse, respectively, and $\lambda_{m1}$ and $\lambda_{m2}$ are the first and second Eigenvalues, respectively.

---

**Algorithm 1** Examining whether a collision occurs, based on the GMM model and motion primitives

---

1:  Generate $M$ GMM & $N$ local trajectories
2:  Discretize local trajectories as $K$ multiple points
3:  **for** $m = 1 : M$ **do**
4:      Obtain mean($\mu_m$) & covariance($\Sigma_m$)
5:      Obtain Eigenvalue($\lambda_m$) & Eigenvector($\nu_m$) of $\Sigma_m$
6:      Perform rotational transformation of multiple points in the global coordinate system
7:      **for** $n = 1 : N$ **do**
8:          **for** $k = 1 : K$ **do**
9:              **if** $f(x_k) = \left( \frac{x_k - \mu_{mx}}{\lambda_{m1}} \right)^2 + \left( \frac{y_k - \mu_{my}}{\lambda_{m2}} \right)^2 \leq s$ **then**
10:                 Delete trajectory
11:             **end if**
12:         **end for**
13:     **end for**
14: **end for**

---

## 4. Collision Avoidance Simulation

### 4.1. Simulation Environment

To verify the proposed algorithm, VRX Simulation, a Gazebo robot operating system (ROS) simulation platform, is used. In general, the ROS is used to develop robot applications, with functionalities of hardware abstraction, low-level device control, message transfer between devices, and library provision. VRX is a type of subpackage of the Gazebo simulation, which corresponds to a virtual ship simulation developed in cooperation with the Naval Postgraduate School, Office of Naval Research and Open Robotics. In the simulation, a wave adaptive modular vessel (WAM-V) is used, and the shape and specifications are presented in Figure 8 and Table 1, respectively. WAM-V is a twin-propelled ship equipped with two thrusters in the stern. It is a differential thruster-type vessel that changes direction by generating a steering control moment caused by the difference in rpm. The steering command is defined as Equations (10) and (11) [24]. $\delta_{mean}$ refers to the average RPM of both thrusters and $\delta_{cmd}$ is the difference in the RPM of the thrusters, which are normalized to have a value between $-1$ and $1$ by divided by the maximum speed. $n_{port}, n_{stbd}, n_{max}$ refer to RPM of the left/right thruster and the maximum RPM of the thruster respectively.

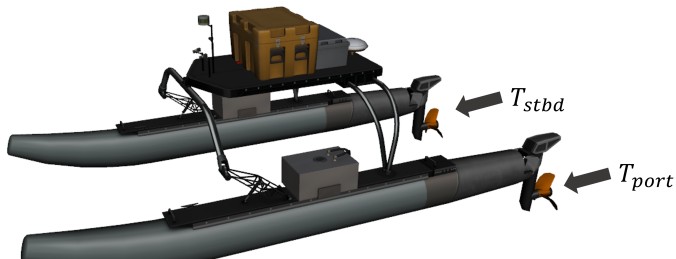

**Figure 8.** Appearance of WAM-V.

**Table 1.** Specifications of WAM-V.

| Parameter | Value |
|:---:|:---:|
| Length | 4.85 m |
| Width | 2.44 m |
| Height | 1.27 m |
| Weight (No load) | 154 kg |
| Payload | 220 kg |
| Full load draft | 0.17 m |
| Operating speed | 2.3 kn |

$$\delta_{mean} = (n_{port} + n_{stbd})/2n_{max} \tag{10}$$

$$\delta_{cmd} = (n_{port} - n_{stbd})/2n_{max} \tag{11}$$

In VRX, Fossen's 6-DOF vectorial model was adopted to express the motion of WAM-V. The equation of motion is expressed in Equation (12) [25].

$$\underbrace{M_{RB}\dot{v} + C_{RB}(v)v}_{rigid\,body\,forces} + \underbrace{M_A\dot{v}_r + C_A(v_r)v_r + D(v_r)v_r}_{hydrodynamic\,forces} + \underbrace{g(\eta)}_{hydrostatic\,forces} = \tau_{propulsion} + \tau_{wind} + \tau_{waves} \tag{12}$$

where

$$\eta = [x, y, z, \phi, \theta, \psi]^T$$
$$v = [u, v, w, p, q, r]^T \tag{13}$$

$\eta$ and $v$ stand for position and velocity vectors for surge, sway, heave, roll, pitch, and yaw, respectively. $v_r$ means the relative speed of the vessel with respect to the fluid. On the right-hand side, the terms mean the forces and moments generated by the thruster, wind, and waves, in order. However, in this study, the effects of wind and waves are not considered, so $\tau_{wind}$ and $\tau_{waves}$ can be omitted. The hydrodynamic force term includes added mass. The hydrodynamic derivatives follow the SNAME (1950) notation. The simplified added mass term is expressed as Equation (14).

$$M_A = - \begin{bmatrix} X_{\dot{u}} & 0 & 0 & 0 & 0 & 0 \\ 0 & Y_{\dot{v}} & 0 & 0 & 0 & Y_{\dot{r}} \\ 0 & 0 & 0 & 0 & 0 & 0 \\ 0 & 0 & 0 & 0 & 0 & 0 \\ 0 & 0 & 0 & 0 & 0 & 0 \\ 0 & N_{\dot{v}} & 0 & 0 & 0 & N_{\dot{r}} \end{bmatrix} \tag{14}$$

where $N_{\dot{v}} = Y_{\dot{r}}$.

The Coriolis-centripetal matrix is expressed as Equation (15).

$$C_A(v_r) = - \begin{bmatrix} 0 & 0 & 0 & 0 & 0 & Y_{\dot{v}}v_r + Y_{\dot{r}}r \\ 0 & 0 & 0 & 0 & 0 & -X_{\dot{u}}u_r \\ 0 & 0 & 0 & 0 & 0 & 0 \\ 0 & 0 & 0 & 0 & 0 & 0 \\ 0 & 0 & 0 & 0 & 0 & 0 \\ 0 & 0 & 0 & -Y_{\dot{v}}v_r - Y_{\dot{r}}r & X_{\dot{u}}u_r & 0 \end{bmatrix} \tag{15}$$

As in Equation (16), the hydrodynamic damping matrix $D(v_r)$ consists of the sum of the linear term $D_l$ and the quadratic term $D_n(v_r)$. Each term is expressed in Equations (17) and (18).

$$D(v_r) = D_l + D_n(v_r) \tag{16}$$

$$D_l = - \begin{bmatrix} X_u & 0 & 0 & 0 & 0 & 0 \\ 0 & Y_v & 0 & 0 & 0 & Y_r \\ 0 & 0 & Z_w & 0 & 0 & 0 \\ 0 & 0 & 0 & K_p & 0 & 0 \\ 0 & 0 & 0 & 0 & M_q & 0 \\ 0 & N_v & 0 & 0 & 0 & N_r \end{bmatrix} \tag{17}$$

$$D_n(v_r) = - \begin{bmatrix} X_{u|u|}|u_r| & 0 & 0 & 0 & 0 & 0 \\ 0 & Y_{v|v|}|v_r| + Y_{|r|v}|r| & 0 & 0 & 0 & Y_{|v|r}|v_r| + Y_{|r|r}|r| \\ 0 & 0 & 0 & 0 & 0 & 0 \\ 0 & 0 & 0 & 0 & 0 & 0 \\ 0 & 0 & 0 & 0 & 0 & 0 \\ 0 & N_{|v|v}|v_r| + N_{|r|v}|r| & 0 & 0 & 0 & N_{|v|r}|v_r| + N_{r|r|}|r| \end{bmatrix} \tag{18}$$

The control method of the WAM-V is as follows. After calculating the desired psi considering the locations of the vessel and target point, the desired yawrate and delta command are sequentially obtained through the psi controller and yawrate controller. Subsequently, the delta command is transmitted to the actuator to drive the thrusters. The controller has a double-loop control structure, and the gains of the controller are calculated through trial and error. The flow of control is presented as a block diagram in Figure 9.

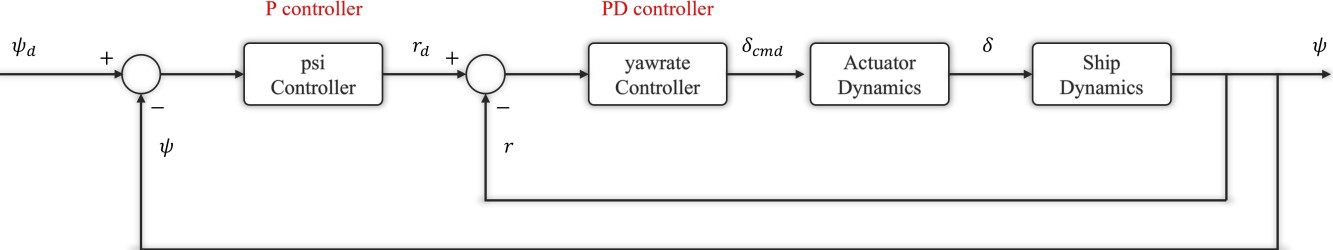

**Figure 9.** Process flow of ship actuator control.

The simulation environment is as follows. Obstacles are placed in the path of the WAM-V, which is set to sail toward a target point. No external force such as winds or currents are introduced. Collision avoidance is considered to fail when a maneuvering vessel and an obstacle come into contact, and the obstacle deviates by a certain distance. To verify the validity of the collision avoidance algorithm, two scenarios are considered.

### 4.1.1. Scenario with Multiple Fixed Obstacles

The objective is to avoid multiple fixed obstacles. Many obstacles were placed at random positions. Figure 10 shows the arrangement of the fixed obstacles and moving direction of the ship. The target point is 150 m ahead of the ship. The obstacles are spherical buoys with a diameter of 1 m, randomly placed in a 110 m × 30 m space on the path.

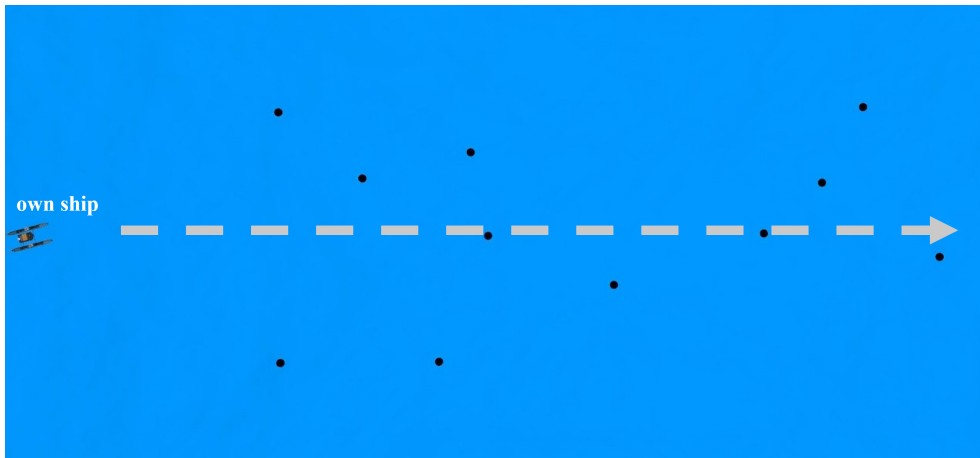

**Figure 10.** Scenario with multiple fixed obstacles.

4.1.2. Scenario with Fixed and Mobile Obstacles

This situation involves multiple obstacles and two vessels whose routes cross with that of the ship. First, an obstacle ship approaches the front of the own vessel. Subsequently, the other obstacle ship approaches from the port side of the own vessel. Figure 11 shows the arrangement of the fixed and moving obstacles and the desired path of the own ship and obstacle ships. The obstacle ships move only in the surge direction at a speed of approximately 1.4 kn (1 kn = 0.51 m/s).

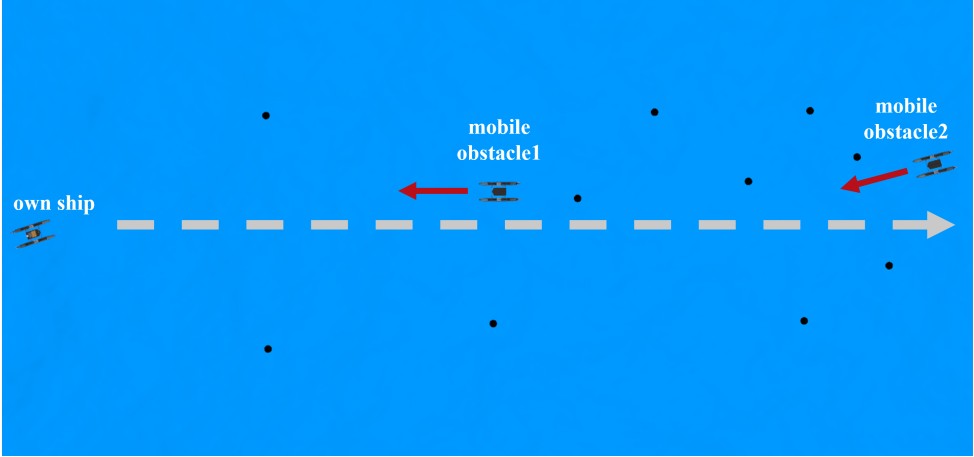

**Figure 11.** Scenario with fixed and mobile obstacles.

The performance of the proposed method based on GMM is compared with that of the method based on the VFF. The success/failure of collision avoidance is evaluated by calculating the minimum distance between the obstacle and the own ship. The sum of half the width of the ship and radius of the obstacle buoy is set as the threshold for collision judgment. A collision is considered to occur when the minimum distance is smaller than the threshold.

*4.2. Simulation Results*

The information obtained from the simulation is visualized through the RViz program. RViz is a 3D visualization tool that can be used to visualize and examine the data in the ROS environment. For example, using the shape of a ship, LiDAR point cloud information, GMM data calculated from obstacles, and trajectories of motion primitives in a virtual space, the progress can be conveniently evaluated. The following subsections describe the results of the simulations.

4.2.1. Scenario with Multiple Fixed Obstacles

Figure 12 shows the simulation screen for the scenario with fixed obstacles. Figure 12a shows the trajectory of the USV toward the target point in intervals of 8 s. The USV successfully arrives at the target point while avoiding the obstacles. Figure 12b shows the virtual information around the ship, expressed in RViz. ①, ②, and ③ in (b) indicate the GMM mapping at ship positions shown in (a). The red dots represent the LiDAR point cloud information and the yellow ellipse is the confidence ellipse determined based on the point cloud information, that is, the obstacle information captured by the USV to avoid collision. The curve located in front of the USV represents the local trajectories generated by the unicycle model. The blue and red curves are trajectories in the direction in which the USV does not collide and does collide with other obstacles, respectively. The vessel controls the direction by adopting the optimal path closest to the desired psi among the trajectories that do not lead to a collision, and the optimal path is displayed as a green curve on RViz. The shortest path to the waypoint is indicated by a pink dotted line. As shown in the bottom part of the figure, a path with a range larger than the actual obstacle

size corresponds to a collision. In other words, a safety margin for the collision prediction is set to ensure the safety of the ship.

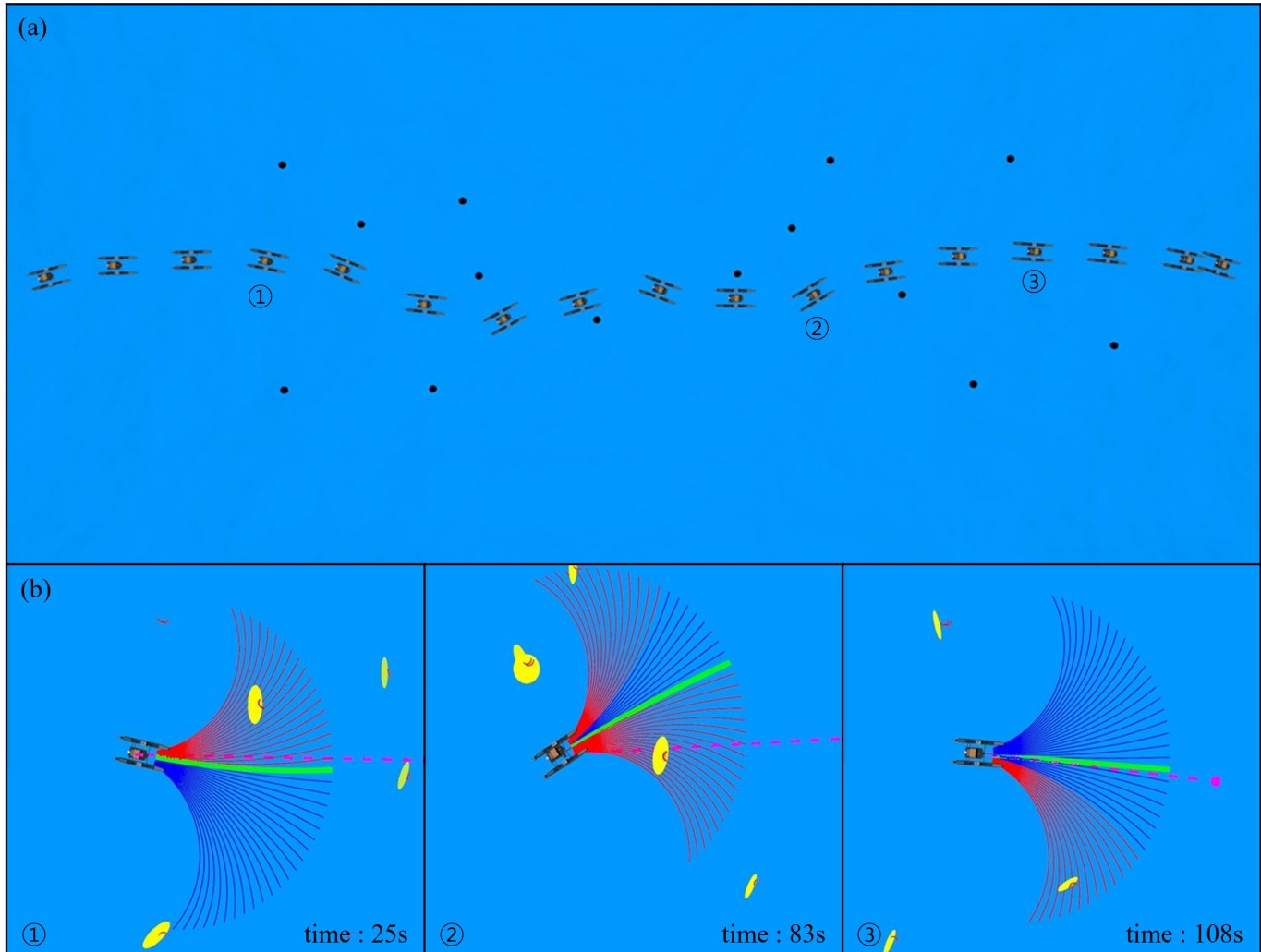

**Figure 12.** Results of ship trajectory and GMM mapping for stationary obstacles.(**a**) Trajectory of the USV expressed in Gazebo; (**b**) Virtual information around the USV expressed in RViz.

Figure 13a shows the state variables of the USV, which vary in the simulation. The horizontal axis t indicates the elapsed time, and the vertical axis of each graph is the state variable of the USV. $u$ represents the surge speed of the ship. The ship progresses while maintaining an operating speed of 2.3 kn. $\psi$ and r denote the heading angle and yawrate of the USV, respectively. The solid red line represents $r_d$, which is the desired yawrate when the optimal path is selected in motion primitives, and the solid blue line is the current yawrate. Except in a certain section, the desired yawrate is suitably followed. $\delta_{cmd}$ is the steering command. The thruster command value is $T_{cmd} \in [0, 1]$. When the USV advances, the left and right thrusters receive a command value of 0.5. While turning hard to the port and starboard sides, the left and right thrusters receive values of 0 and 1 and values of 1 and 0, respectively. Therefore, a positive and negative delta command is transmitted when turning left and right, respectively. Figure 13b shows the distance to the closest obstacle to determine whether the USV collides with the obstacle. $d_{obs}$ is the distance between the USV and the nearest obstacle, corresponding to the Euclidean distance. $d_{obs}$ is defined as in Equation (19), where $(x_{obs}, y_{obs})$ are the obstacle coordinates.

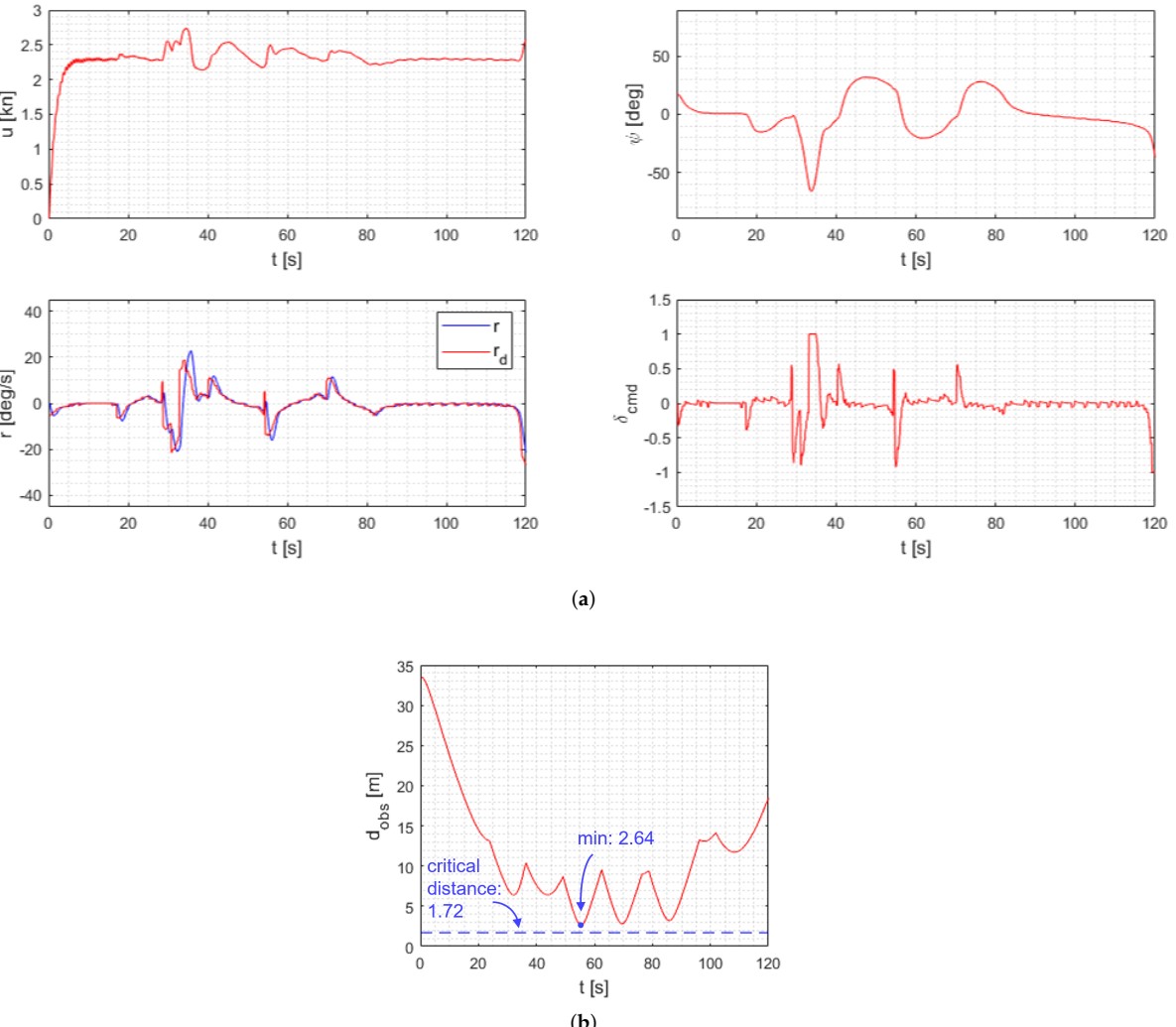

**Figure 13.** Variables measured in the scenario with multiple fixed obstacles. (**a**) State variables of the USV; (**b**) distance to the closest obstacle.

$$d_{obs} = min(|x_{os} - x_{obs}| + |y_{os} - y_{obs}|) \tag{19}$$

In Figure 13b, the critical distance at which the ship collides with the obstacles is 1.72 m and the minimum distance between the USV and the obstacle is 2.64 m. Therefore, the USV can proceed toward the destination without any collision.

Next, to verify the effectiveness of the GMM·MP-based collision avoidance method, it is compared with the VFF theory. Figure 14 shows representative trajectories of GMM·MP method and VFF method for fixed obstacles. The simulation for each method was performed 10 times, and the average values of the results presented in Table 2.

The finish time is the time elapsed from the initial movement of the unmanned ship to when it is 5 m from the target point. The finish time of the GMM·MP method is 3.83 s smaller than that of the VFF method. On average, the distance from the nearest obstacle for the GMM·MP method is 0.02 m smaller than that for the VFF method. The GMM·MP method and VFF method can successfully avoid collision with obstacles, as indicated by the number of collisions and percentage of success (number of simulations completed without any collisions). The cross-track error indicates the degree of deviation of the USV from the virtual straight line connecting the starting and destination points. The error of the GMM·MP method is 2.74 m smaller than that for the VFF method. To provide the information more intuitively, the deviation of the avoidance performance of

the two collision avoidance methods is expressed as a box plot in Figure 15. $T_f$ is the finish time of the USV, and $E_{crs}$ is the cross-track error. In the first graph, although the standard deviation of $T_f$ for the GMM·MP method is greater than that for the VFF method, the GMM·MP method exhibits a smaller finish time than the VFF method in all simulations. For the GMM·MP method, the maximum and minimum finish time are 127.92 and 126.18 s, respectively. The corresponding values for the VFF method are 131.14 s and 130.66 s. As shown in the second graph, the distance from the obstacle is smaller for the GMM·MP method than that for the VFF method. The standard deviation for the GMM·MP method and the VFF method are 0.12 m and 0.07 m, respectively. The VFF method yields more stable results, likely because the vessel in this case does not pass through the narrow space between the obstacles. The standard deviation of the cross-track error is 0.06 m for the GMM·MP method, corresponding to a higher stability than that for the VFF method (0.1 m).

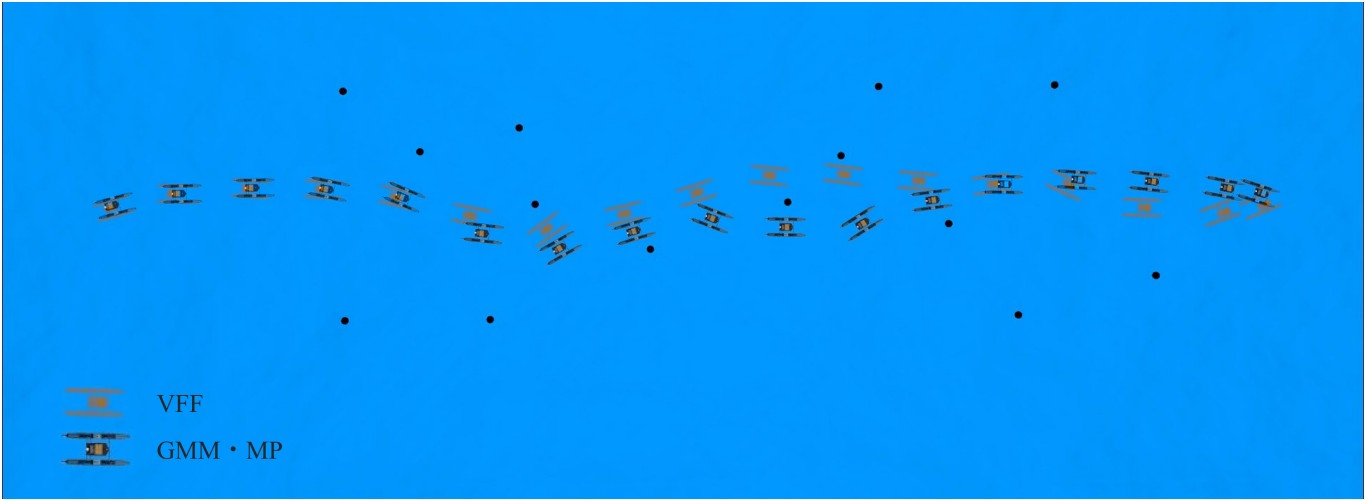

**Figure 14.** Comparison of GMM·MP and VFF trajectories for fixed obstacles.

**Table 2.** Evaluation of obstacle avoidance performance for different collision avoidance methods in the scenario with multiple fixed obstacles.

|  | GMM·MP (Proposed) | VFF |
|---|---|---|
| Finish time | 127.07 s | 130.9 s |
| Distance from nearest obstacle | 9.76 m | 9.78 m |
| Number of collisions | 0 | 0 |
| Percentage of success | 100% | 100% |
| Cross-track error | 2.28 m | 5.02 m |

Number of trials: 10.

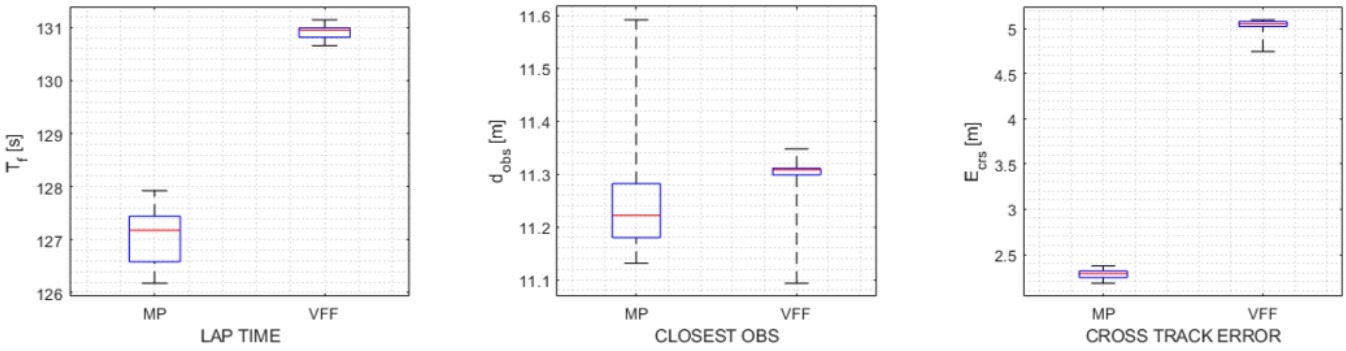

**Figure 15.** Avoidance performance of the USV in the scenario with multiple fixed obstacles.

#### 4.2.2. Scenario with Fixed and Mobile Obstacles

Figure 16 shows the trajectory for the case in which the USV performs collision avoidance for fixed and mobile obstacles, in intervals of 8 s. The simulation environment is shown in Figure 16a, and the GMM mapping and local trajectories are shown in Figure 16b. In (a), the path of the target ships is marked by faint lines. The point at which the ship starts to avoid the target ships is highlighted in orange. ①, ②, and ③ in the (b) represent the results of GMM mapping of obstacles at positions shown in (a). As indicated in (a), the ship first encounters an obstacle ship at point ① in a head-on situation and turns to starboard. At point ②, the ship encounters the second obstacle ship in the crossing situation, and thus, collision avoidance is implemented. At point ③, collision avoidance for fixed obstacles is performed.

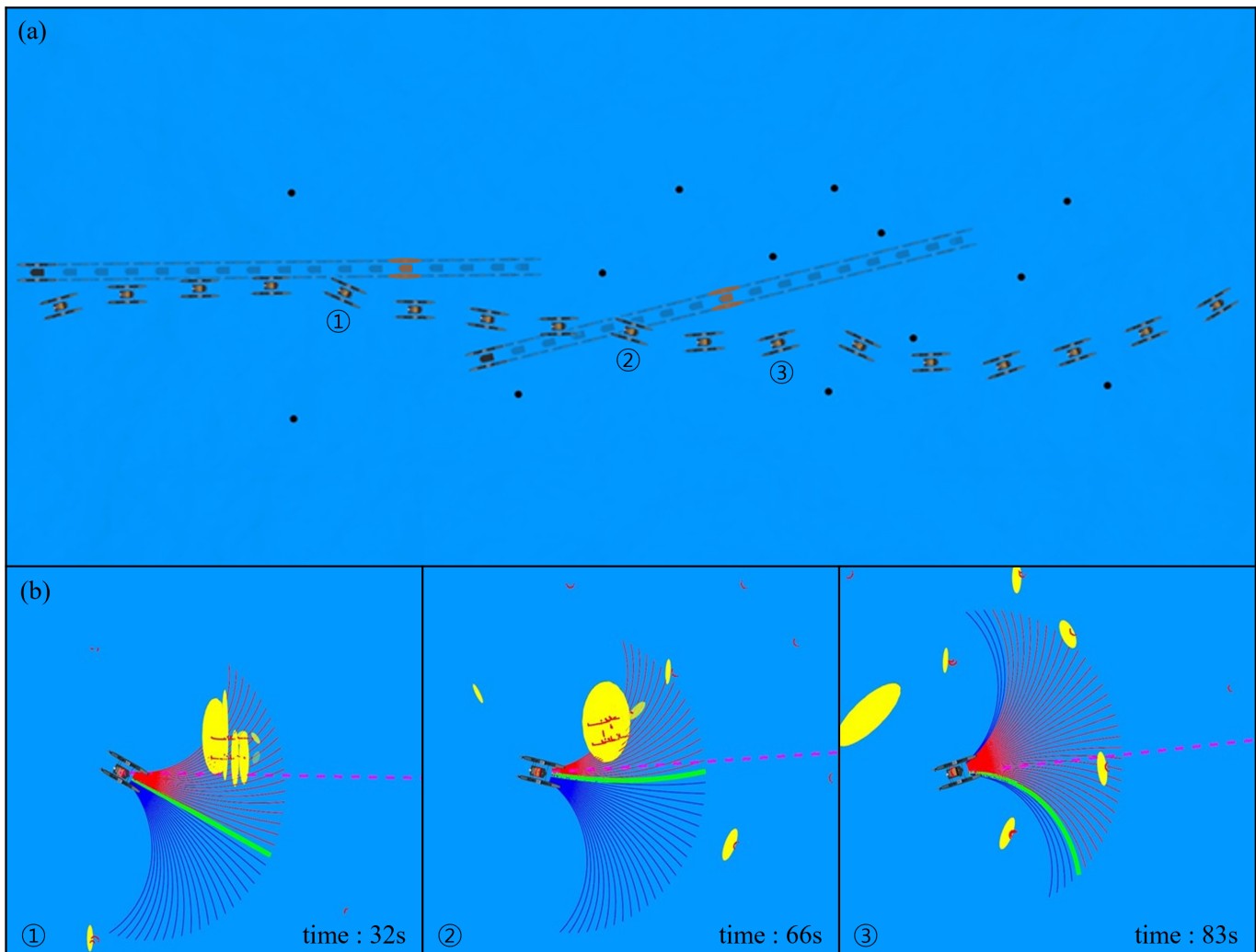

**Figure 16.** Results of ship trajectory and GMM mapping in the scenario with fixed and mobile obstacles. (**a**) Trajectory of the USV expressed in Gazebo; (**b**) Virtual information around the USV expressed in RViz.

Figure 17 shows the state variables of the USV in the scenario with fixed and mobile obstacles. Figure 17a shows the state variables of the USV during collision avoidance in the form of a graph. The first graph shows that the average ship speed is maintained at 2.3 kn. The ship speed rapidly increases after 70–80 s. This phenomenon occurs because $\delta_{cmd}$ is maximized while avoiding dynamic obstacles. Moreover, as shown in the third graph, the yawrate is effectively tracked. However, the control response becomes slower in the section in which the desired yawrate rapidly changes. Figure 17b shows the distance between

the USV and nearest obstacle. The minimum distance is 2.36 m, which is greater than the collision distance threshold (1.72 m).

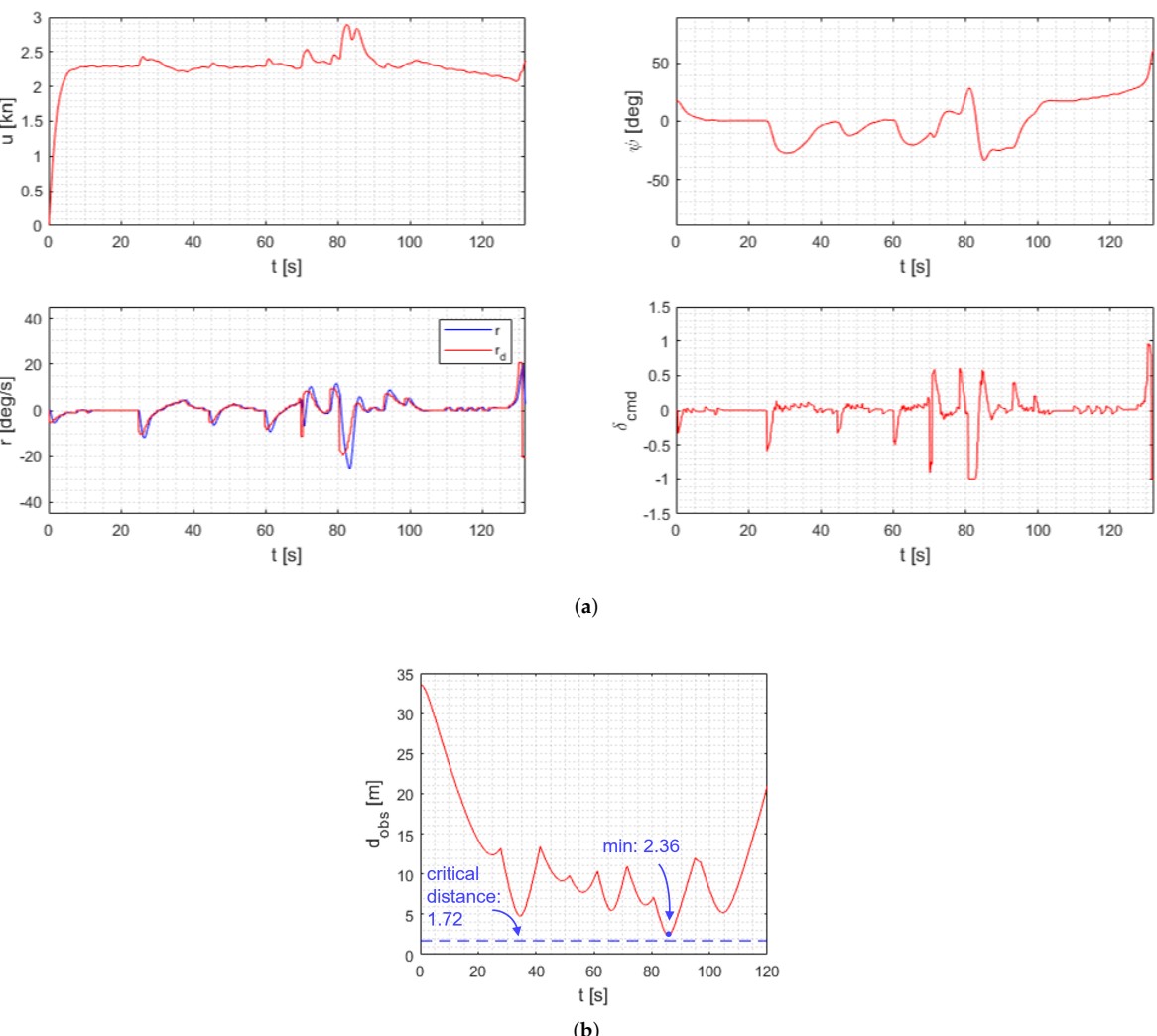

**Figure 17.** Variables in the scenario with fixed and mobile obstacles. (**a**) State variables of the USV. (**b**) Distance to the closest obstacle.

Ten simulations are performed to verify the collision avoidance performance of the USV against dynamic obstacles. Figure 18 shows representative trajectories of GMM·MP method and VFF method in mobile obstacle scenario. Table 3 summarizes the results of the different collision avoidance methods for dynamic obstacles. The finish time for the GMM·MP method is approximately 2.67 s smaller than that of the VFF method, although the distance from the nearest obstacle is approximately 0.32 m larger for the VFF method. Using the GMM·MP method, all 10 simulations were successfully completed without collisions. However, when using the VFF method, collisions occurred once in two simulations.

Figure 19 shows the deviation in the avoidance performance of the USV for dynamic obstacles as a box plot. The standard deviation of the finish time is 0.37 s and 0.55 s for the GMM·MP method and VFF method, respectively. Therefore, the GMM·MP method exhibits a fast and stable response. The distance to the nearest obstacle is 0.54 m and 0.42 m for the GMM·MP and VFF methods, respectively. Therefore, the VFF method is slightly more stable. However, the standard deviations of the cross-track error are 0.08 m and 0.61 m for the GMM·MP and VFF, respectively. Therefore, the GMM method is more stable. This finding can be explained by the fact that a collision occurs when using the VFF method, and an outlier exists in the cross-track error.

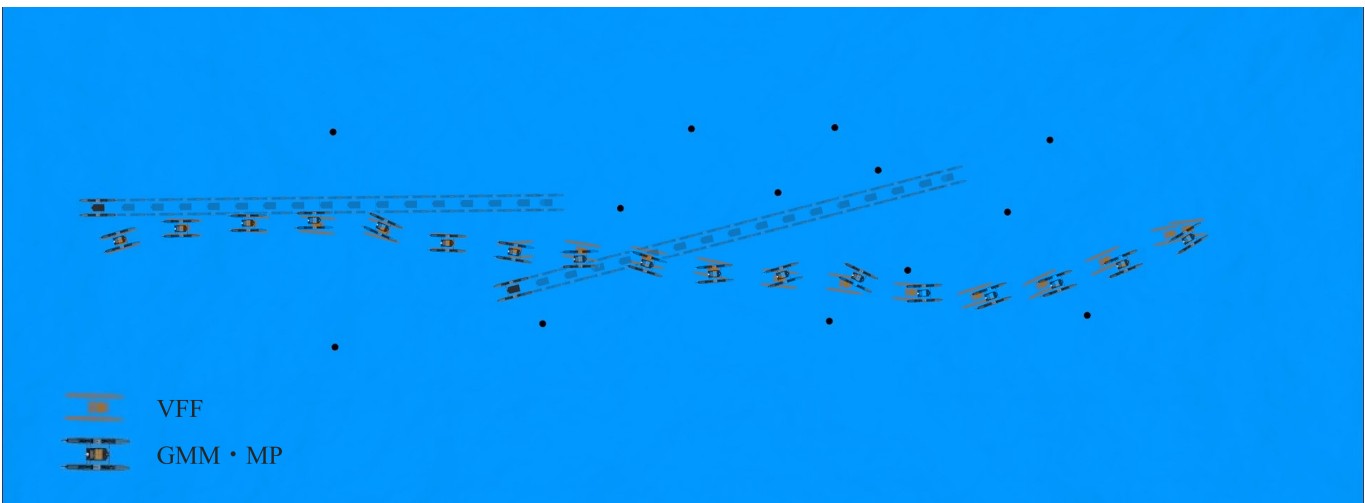

**Figure 18.** Comparison of GMM·MP and VFF trajectories for mobile obstacles.

**Table 3.** Evaluation of obstacle avoidance performance of different collision avoidance methods in the scenario with fixed and mobile obstacles.

|  | **GMM·MP (Proposed)** | **VFF** |
|---|---|---|
| Finish time | 132.74 s | 135.41 s |
| Distance from nearest obstacle | 9.55 m | 9.87 m |
| Number of collision | 0 | 2 |
| Percentage of success | 100% | 80% |
| Cross-track error | 3.69 m | 3.58 m |

Number of trials: 10.

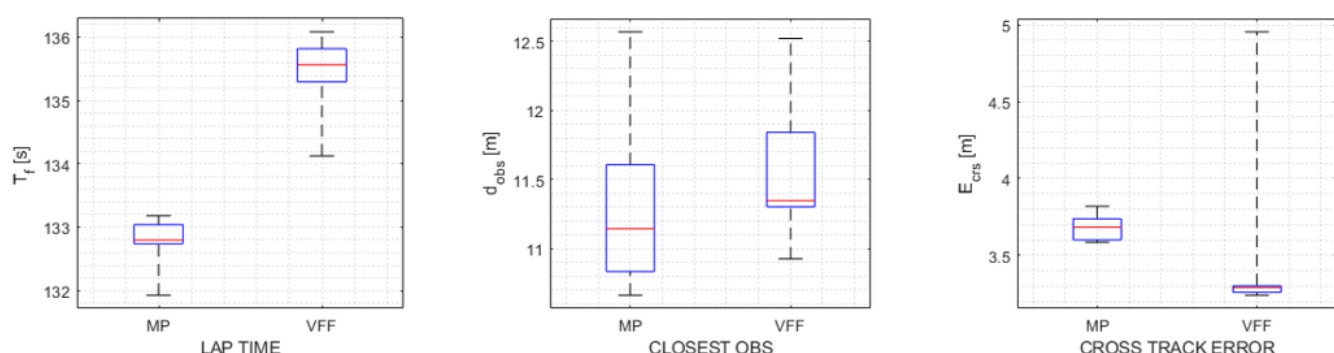

**Figure 19.** Avoidance performance of the USV in the scenario with multiple fixed and mobile obstacles.

### 4.3. Overview of the Results

The GMM·MP generally outperforms the VFF method. When the VFF theory is applied, the vessel tends to avoid obstacles by taking a detour without passing through the narrow space between obstacles. In contrast, the GMM·MP method allows the vessel to pass through narrow passages. Nevertheless, because the GMM·MP method implements obstacle mapping based on probability, obstacle mapping is performed differently even in the same simulation environment, and thus, the standard deviation of the avoidance performance is larger than that associated with the VFF method. In terms of the maintenance of distance from the obstacles, the VFF method exhibits more stable results.

## 5. Conclusions

According to collision avoidance simulations, the vessel implementing the proposed algorithm can arrive at the target point without colliding with an obstacle. Notably, the method based on GMM mapping and the unicycle model does not yield the exact position

of obstacles or vessel behavior compared to the model that considers dynamics. However, using the GMM, the location of the obstacles can be approximated, and the trajectory of the vessel can be corrected through prompt calculation to avoid a collision situation. In addition, the stability of the proposed method was verified by confirming that the USV can safely avoid obstacles in ROS simulations. The proposed technique, which can correct the local trajectory according to obstacles while maintaining the current route, is expected to enhance the reliability of path planning and collision avoidance when used in combination with existing deliberative control methods. Finally, this technology is expected to be useful when performing the USV formation control by preventing collision close USVs or obstacles while maintaining their formation.

**Author Contributions:** Conceptualization, D.L. and J.W.; methodology, D.L. and J.W.; software, D.L.; validation, J.W.; formal analysis, D.L. and J.W.; investigation, D.L.; resources, D.L. and J.W.; data curation, D.L.; writing—original draft preparation, D.L.; writing—review and editing, D.L.; visualization, D.L.; supervision, J.W.; project administration, J.W.; funding acquisition, J.W. All authors have read and agreed to the published version of the manuscript.

**Funding:** This work was supported by the National Research Foundation of Korea(NRF) grant funded by the Korea government(MSIT) (No. 2021R1G1A1095671).

**Institutional Review Board Statement:** Not applicable.

**Informed Consent Statement:** Not applicable.

**Data Availability Statement:** The data that support the findings of this study are available from the corresponding author upon reasonable request.

**Conflicts of Interest:** The authors declare no conflict of interest.

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
