# Peer review of "Reactive Collision Avoidance of an Unmanned Surface Vehicle through Gaussian Mixture Model-Based Online Mapping"

_jmse, doi:10.3390/jmse10040472_

Round 1

Reviewer 1 Report

This paper proposes a reactive collision avoidance algorithm to ensure the safety of USVs against obstacles. From a technical point of view, there are still some problems in this paper that need to be revised. There are some remarks and comments which should be addressed to:

Comment 1: Please specify the reasons for choosing GMM method for ship collision avoidance research. In my opinion, now scholars prefer to study the application of intelligent algorithm in USV path planning. Please give the analysis of advantages and disadvantages.

Comment 2: The motion state of the ship at sea is six degrees of freedom, and this paper uses the unicycle model to predict the ship behavior. Please explain the feasibility between the two in combination with relevant kinematic equations.

Comment 3: I notice that the dynamic collision avoidance rules proposed in this paper are set by the author himself. And the way of encounter during the experiment is frontal encounter. Has the author considered the pursuit problem? Have you referred to COLREGs? Why not design collision avoidance rules based on COLREGs?

Comment 4: In the simulation results, you only compared with VFF. It is suggested to supplement more comparisons with the latest algorithms to enhance the persuasiveness of the article.

Comment 5: You could research USV formation or distributed USV. It has little practical significance for a single USV in theory.

Author Response

Dear Reviewer 1,

Thank you so much for the review. I'm writing to submit my revision file.

I've read your review carefully and made some corrections.

I look forward to receiving your reply.

Best regards,

Dongwoo Lee

Reviewer 2 Report

The article mainly works onreactivecollisionavoidance method.The GMM·MP proposed by the authors may be a simple and efficient collision avoidance method in USV navigation. I will appreciate it if the authors could address the following questions and concerns:

  1. The authors are suggested to enhance the quality of the figures, like Figure 3 and Figure 4 etc.
  2. The authors are suggested to check the pseudocode writing format of the algorithm. Symbols such as for and if need to be bolded.
  3. The number of the formula in line 112 of the manuscript is wrong. The authors are suggested to check whether the numbers of the formulas and figures match the content of the manuscript correctly.
  4. The author uses VFF as the comparison of GMM·MP, and the experimental results are listed in Table 2 and Table 3. However, readers would like to see the trajectory comparison of GMM·MP and VFF in collision avoidance tests. Therefore, the authors are suggested to show the navigation trajectories of other methods in the experimental results.
  5. Could the authors provide another method other than virtual force field to compare with your proposed GMMMP?
  6. Figure 15(a) doesn’t show the distance between the USV and the nearest obstacle. It is suggested to add the distance.

Author Response

Dear Reviewer 2,

Thank you so much for the review. I'm writing to submit my revision file.

I've read your review carefully and made some corrections.

I look forward to receiving your reply.

Best regards,

Dongwoo Lee

Reviewer 3 Report

The proposed algorithm of GMM.MP-based collision avoidance method is to compare to VFF theory. In my opinion, this paper presents several scenarios of simulation analysis but lacks a mathematical model of ship motion and WAM-V parameters of hydrodynamic. The reviewer thinks that GMM mapping and the unicycle model can approximate the location of obstacles.

This manuscript is an exciting topic, however, the effectiveness of the proposed algorithm is unclear. Here are the comments.

In section 2.1
-In line 82, the expectation-maximization (EM) algorithm should be added to the related papers.
-The authors should explain more detail the parameter in Eq.2

In section 2.2
-Figure 2 should explain: lamda & s 

In section 3.1
-In line 138, what is the pure pursuit algorithm? The authors should explain more.

In section 3.2
-Eq.7 should be checked. The reviewer think that xt+T = xT + ....

Section 4.1 Simulation environment

- In table 1, the essential dimension must include the center of gravity, the center of buoyancy, draft, and trim at bow and stern. This is because the ship motion equation is related to the WAM-V parameters.
- In Fig. 10, the scenario with fixed and mobile obstacles, the maneuvering of WAM-V is not demonstrated. The proposed algorithm works well, but the paper does not demonstrate the WAM-V with ruder system validation.
- In line 251, the critical distance is 1.72m. These are the simulation results. However, as an above comment, the reviewer does not think the maneuvering of WAM-V shows the same results.

Author Response

Dear Reviewer 3,

Thank you so much for the review. I'm writing to submit my revision file.

I've read your review carefully and made some corrections.

I look forward to receiving your reply.

Best regards,

Dongwoo Lee

Round 2

Reviewer 1 Report

Thank you very much for your reply.I think this article can be published after your modification.

Reviewer 2 Report

I have no further comments.

If the authors have enough time, the raised comment that remained unchanged helps.

Reviewer 3 Report

The authors have carefully considered the comments and made corrections to the manuscript. I would suggest that this manuscript is acceptable for further procedure of publication.